# Evaluating the Risk of Inguinal Lymph Node Metastases before Surgery Using the Morphonode Predictive Model: A Prospective Diagnostic Study in Vulvar Cancer Patients

**DOI:** 10.3390/cancers15041121

**Published:** 2023-02-09

**Authors:** Simona Maria Fragomeni, Francesca Moro, Fernando Palluzzi, Floriana Mascilini, Vittoria Rufini, Angela Collarino, Frediano Inzani, Luciano Giacò, Giovanni Scambia, Antonia Carla Testa, Giorgia Garganese

**Affiliations:** 1Unità di Ginecologia Oncologica, Dipartimento Scienze della Salute della Donna, del Bambino e di Sanità Pubblica, Fondazione Policlinico Universitario A. Gemelli IRCCS, Largo A. Gemelli, 8, 00168 Rome, Italy; 2Bioinformatics Facility, Gemelli Science and Technology Park (GSTeP), Fondazione Policlinico Universitario A. Gemelli IRCCS, Largo A. Gemelli, 8, 00168 Rome, Italy; 3Section of Nuclear Medicine, Department of Radiological Sciences and Haematology, Università Cattolica del Sacro Cuore, Largo A. Gemelli, 8, 00168 Rome, Italy; 4Unit of Nuclear Medicine, Fondazione Policlinico Universitario A. Gemelli IRCCS, Largo A. Gemelli, 8, 00168 Rome, Italy; 5Anatomic Pathology Unit, Department of Molecular Medicine, University of Pavia, Viale C. Golgi 19, 27100 Pavia, Italy; 6Dipartimento Universitario Scienze della Vita e Sanità Pubblica, Sezione di Ginecologia ed Ostetricia, Università Cattolica del Sacro Cuore, Largo A. Gemelli, 8, 00168 Rome, Italy

**Keywords:** ultrasound, machine learning, lymph nodes, vulvar cancer

## Abstract

**Simple Summary:**

Inguinal node status represents one of the key elements in defining prognosis and treatment strategies in vulvar cancer patients. Preoperative lymph node staging is still a challenging topic. Several imaging methods are currently recommended in the guidelines (CT, PET/CT, MRI, US) based on performance data that are still not conclusive in the literature. Recently, ultrasound is emerging as the method of choice for preoperative evaluation of the inguinofemoral LN, but only when performed by experienced operators, given the limited reliability of subjective evaluation by unskilled operators. The morphonode predictive model represents an artificial intelligence tool that aims to overcome this limitation by supporting the standard ultrasound in adequately predicting the presence of lymph node metastases for improving preoperative surgical planning. We plan to proceed with further multicenter prospective validation and further develop the actual model, including both clinical and biological data.

**Abstract:**

Ultrasound examination is an accurate method in the preoperative evaluation of the inguinofemoral lymph nodes when performed by experienced operators. The purpose of the study was to build a robust, multi-modular model based on machine learning to discriminate between metastatic and non-metastatic inguinal lymph nodes in patients with vulvar cancer. One hundred and twenty-seven women were selected at our center from March 2017 to April 2020, and 237 inguinal regions were analyzed (75 were metastatic and 162 were non-metastatic at histology). Ultrasound was performed before surgery by experienced examiners. Ultrasound features were defined according to previous studies and collected prospectively. Fourteen informative features were used to train and test the machine to obtain a diagnostic model (Morphonode Predictive Model). The following data classifiers were integrated: (I) random forest classifiers (RCF), (II) regression binomial model (RBM), (III) decisional tree (DT), and (IV) similarity profiling (SP). RFC predicted metastatic/non-metastatic lymph nodes with an accuracy of 93.3% and a negative predictive value of 97.1%. DT identified four specific signatures correlated with the risk of metastases and the point risk of each signature was 100%, 81%, 16% and 4%, respectively. The Morphonode Predictive Model could be easily integrated into the clinical routine for preoperative stratification of vulvar cancer patients.

## 1. Introduction

Preoperative assessment of the clinical stage in vulvar cancer patients represents a relevant challenge. Several imaging methods are currently in use to stage vulvar cancer patients, but as yet, there is no defined gold standard. Computed Tomography (CT) scan and Fluorodeoxyglucose-Positron Emission Tomography/CT (^18F^FDG-PET/CT) are commonly used to assess primary tumors and distant metastases. In particular, in the detection of inguinal lymph node metastases, CT scans have shown a negative predictive value of around 75%, while ^18F^FDG-PET/CT has recently been reported to exceed 90% [1,2,3,4,5,6,7]. Magnetic Resonance Imaging (MRI) is another technique useful to assess the extension of disease [2,3], and this method has shown sensitivity and specificity ranging from 40% to 89% and 81% to 100%, respectively, in detecting lymph node metastasis [4].

Since 2017, international guidelines [5] have also mentioned ultrasound in the management of patients with vulvar cancer, especially to guide needle aspiration or core biopsy of suspected inguinal lymph nodes.

Currently, FDG-PET-CT and MRI are considered alternatives to groin ultrasound or complementary in cases of suspected distant metastasis [2,3,4,6,7,8,9]. Therefore, inguinal ultrasound should be the first choice when vulvar cancer is diagnosed.

In a previous retrospective study, the Morphonode study, we evaluated the accuracy of ultrasound examination in discriminating between negative and metastatic inguinal lymph nodes in patients with vulvar cancer. All the ultrasound examinations were performed by experts. We demonstrated that all the investigated dimensional and morphological parameters differed significantly between patients with nodal metastases and those with negative lymph nodes. The overall accuracy of subjective assessment at ultrasound was 79%, with a negative predictive value (NPV) of 92% [10]. Moreover, a recent meta-analysis on the role of ultrasound in evaluating inguinal lymph nodes in patients with vulvar cancer confirmed these results, reporting an accuracy of 85% and an NPV of 92%, despite the high heterogeneity in the ultrasound methodology described in the literature [11].

Meanwhile, a multicenter Vulvar International Tumor Analysis (VITA) group has published an international consensus defining standardized terminology to describe the ultrasound characteristics of normal and metastatic lymph nodes and the methodology of performing ultrasound examination [12].

Nevertheless, the accuracy of ultrasound in detecting metastatic inguinal lymph nodes in vulvar cancer patients is still related to the experience of the examiner. No study has currently described the results of this technique when performed regardless of the examiners’ experience, and no supporting diagnostic model has ever been described to identify metastatic lymph nodes. The aim of the present study was to develop a diagnostic model based on machine learning, to discriminate between metastatic and non-metastatic inguinal lymph nodes.

## 2. Materials and Methods 

This is a single-institution prospective diagnostic study including patients enrolled over 3 years (from March 2017 to April 2020) at Gynecologic Oncology Unit, Fondazione Policlinico Universitario A. Gemelli IRCSS. The study was approved by the ethical committee of our University, and signed informed consent was obtained from all patients. All women with histological diagnoses of invasive vulvar cancer (primary tumor or recurrence) who were candidates for inguinal surgery were included in the study. The exclusion criteria were: time from ultrasound evaluation to surgery more than 30 days, the surgical procedure was not performed at our institution, and incomplete clinical and ultrasound data.

Study data were managed using REDCap (electronic data capture tools hosted at Fondazione Policlinico Universitario “A. Gemelli”, IRCCS; https://redcap-irccs.policlinicogemelli.it/, accessed on 8 October 2020).

### 2.1. Ultrasound

All patients underwent preoperative ultrasound examination to assess inguinal lymph nodes and ultrasound-guided fine-needle aspiration cytology or biopsy (FNA) in cases of suspected metastases.

Ultrasound examinations were conducted by two fully-trained gynecologists with more than ten years of experience in the management of vulvar cancer patients (GG and SMF). All ultrasound examinations were performed with MyLab Twice (Esaote, Genova, Italy) machine with 7–12 MHz linear probes.

All the ultrasound parameters of inguinal lymph nodes were prospectively collected using a predefined electronic form (Figure 1), including both the parameters of the “Morphonode study” and the relevant, additional vascular parameters. We chose to rename some of the parameters according to the corresponding current nomenclature of the international consensus on terms and definitions published by the VITA group (Figure 1) [10,12].

The subjective assessment of ultrasound variables was applied to all lymph nodes of each groin by using the following classification in five categories: normal (LN1), reactive but negative (LN2), minimally suspicious (LN3), moderately suspicious (LN4), highly suspicious or positive (LN5). Classes LN3–5 were considered positive (potentially metastatic), while classes LN1–2 were considered negative (non-metastatic). If at least one lymph node was classified as LN3–5, the groin was considered positive overall.

### 2.2. Standard of Reference

All patients underwent surgery on the primary tumor and inguinal lymph nodes, according to international guidelines [5,13]. Lymph node surgery was determined according to the distance between the primary lesion and the midline of the vulva, unilaterally when the distance was greater than 2 cm or bilaterally when less. Sentinel node biopsy (SNB) was performed in cases of primary unifocal tumors less than 4 cm. Patients not satisfying standard criteria for SNB underwent radical inguinal lymphadenectomy. Lymph node histology was considered the reference standard. The tumors were classified according to the revised International Federation of Gynecology and Obstetrics (FIGO) staging system of 2009 [14] that was in effect during the study.

### 2.3. Data Analysis and Sample Size 

#### 2.3.1. Sample Size

The definition of the sample size was not quantifiable as no previous studies had analyzed clinical and ultrasound parameters or developed machine learning models similar to ours. Thus, we set a period of three years to collect all consecutive cases referred to our high-flow institution. In three years, 237 cases were selected. They were included in “training + testing” (4/5 of the total) and “validation” (1/5 of the total) datasets to reduce classification biases and predictive errors caused by the initial selection of cases for model training (i.e., model overfitting problems).

The model—Morphonode predictive model—was developed with the following modular structures: 1. Random forest classifiers, 2. Robust regression binomial model, 3. Decisional tree 4, Similarity profiling. The advantage of this modular architecture was to have a prediction deriving from an ensemble method. These methods are commonly used in machine learning to reduce the prediction error derived from a single classifier.

#### 2.3.2. Malignancy Prediction through Random Forest Classifiers (RFCs): Morphonode–RFC

Based on the observed number of metastatic lymph nodes (*m*) per sample, 237 groins were diagnosed as either *metastatic* (*m* > 0, *n_1_* = 75) or *non-metastatic* (*m* = 0, *n_0_* = 162). Fourteen ultrasound features were selected to train a series of RFCs and generate a feature ranking score based on the importance of each variable in the classification process (Appendix B).

The initial set of samples (*n* = 237) was divided into 5 random subsets, of which four were used in a four-fold cross-validation (i.e., training, testing), and the fifth subset was used as an independent validation set, according to a nested cross-validation procedure [15]. Following this nested procedure [15], the Morphonode–RFC module provides a binary output supported by five RFC classifiers, each one built on four to five of the random subsets used in the validation step. As a new case is provided, five independent predictions are made, and the majority vote determines the final assessment for that case.

Eight indices were used to evaluate RFC performances: sensitivity, specificity, Positive Likelihood Ratio, Negative Likelihood Ratio, Positive Predictive Value (PPV), F1 score (corresponding to the harmonic mean of Sensitivity and Positive Predictive Value), Negative Predictive Value (NPV), accuracy. The RFCs were developed using the R package randomForest [16].

Subjective assessment performances were then evaluated using the same eight performance indices used for RFCs to directly compare the subjective assessment against the Morphonode–RFC (Figure 2).

#### 2.3.3. Malignancy Risk Evaluation: Morphonode–RBM

In order to assess the malignancy risk (*r*) associated with a given ultrasound profile, a robust regression binomial model (RBM) was fitted, considering the 237 samples and the 14 ultrasound variables as predictors (see Appendix B, section “Dichotomization”). This method complements the Morphonode–RFC.

We introduced the F1 score measure to have a threshold less dependent on the possible discrepancy between sensitivity and specificity measures. After estimating the malignancy risk for each groin, two cut-offs were calculated: the one maximizing the F1 score (23%) and the one maximizing both sensitivity and specificity (29%), with respect to the dichotomous outcome metastatic/non-metastatic. This allows us to discriminate between low and high-risk profiles with much higher predictive accuracy. Therefore, groins were stratified as low (<23%), moderate (between 23% and 29%), and high (>29%) malignancy risk. The RBM fitting was performed with the R base function glm, while the bootstrap procedure was implemented using the mosaic [17] R package.

#### 2.3.4. Decision Tree (DT) and Malignancy Signatures: Morphonode–DT

The Morphonode–DT was designed to guide the risk prediction of lymph node metastasis, highlighting the presence of malignant subgroups of patients associated with different ultrasound features. Firstly, ultrasound variables were ranked on the base of their associated malignancy risk *p*, estimated as the positive predictive value of that feature. Features with *p* ≥ 0.85 (i.e., the *p* upper whisker) were considered metastatic markers, directly leading to a malignant diagnosis. Data entries not showing metastatic markers (*p* < 0.85) were partitioned through a decision tree by choosing the best splitting variable, evaluated through Gini impurity. This procedure selected a core of predictive variables. The decision tree was built using the R package rpart [18].

#### 2.3.5. Prediction Error

To assess the reliability of the whole prediction, we implemented a loss function-based error estimate. Given the unavailability of an expected outcome for a new subject, the error *E* of a new prediction is based on a parametric linear relationship between the value of the logistic loss function *L* and the observed error over the available dataset. The parametric function is defined as *E* = *b0* + *bL*. The *b0* value (baseline error term) depends on the metastatic risk signature and reflects the uncertainty deriving from belonging to a specific metastatic risk group. The logistic loss function *L* measures the uncertainty for a single subject as *L* = −1(*y* log(*p*) + (1 − *y*) log(1 − *p*)), where *y* is the predicted RFC outcome, and *p* is the predicted RBM risk estimate. The higher the divergence between RFC and RBM predictions, the higher the uncertainty and so the value of *L*. The *b* coefficient is a correction term accounting for the divergence between the observed subject-level error estimate (calculated as Brier score) and the observed *L* value, both calculated on the available dataset of 237 groins.

#### 2.3.6. Similarity Profiling: Morphonode–SP

As further support to the decisional process, the input ultrasound profile is compared to the ones included in the Morphonode dataset by computing the pairwise similarity between them. The top five most similar profiles are returned to compare the predicted phenotype and ultrasound features with the ones of known similar subjects. The number of top-similar profiles as well as the similarity/correlation functions, can be set to custom values. By default, cosine similarity is used.

#### 2.3.7. Morphonode Predictive Model Implementation

The Morphonode Predictive Model was implemented in the open-source R package Morphonode Predictive Model, available at: https://github.com/Morphonodepredictivemodel, accessed on 23 November 2022.

The package is structured in two simple steps: the input of the data and the predictive output. The output included: dichotomous classification through RFCs (Morphonode–RFC), point malignancy risk prediction through RBM (Morphonode–RBM), malignancy risk signature generated by the DT (Morphonode–DT), and the top five similar profiles (Morphonode–SP).

## 3. Results

### 3.1. Study Population

A total of 324 patients affected by vulvar carcinoma were treated at our institution from March 2017 to April 2020, and 127 of them were eligible for this study and included in the final analysis: 71 patients had negative inguinofemoral lymph nodes at histology (N0 group) while 56 patients were positive for metastases (N1 group) (Figure 3). A total of 237 groins were analyzed, including 162 groins negative for disease and 75 with at least one metastatic lymph node.

### 3.2. Clinical, Surgical, Histopathologic and Ultrasound Features

Clinical, surgical and histopathologic features of the study population are shown in Table 1. The median age at diagnosis was 69 years (range 32–95), with no statistical differences between the two groups (N0 group vs. N1 group). Maximum tumor diameter significantly differed between the two groups (median tumor diameter of 41 mm in the N1 group vs. 25.5 mm in the N0 group, *p*-value < 0.001).

Most patients were FIGO stage I (59, 46.5%), 4 (3.2%) were FIGO stage II, 40 (31.5%) were FIGO stage III and 4 (3.2%) were FIGO stage IV. Twelve patients (9.4%) were affected by disease relapse, and eight cases (6.3%) were evaluated after primary chemo-radiation (RT/CT). The most frequent histotype was invasive squamous carcinoma (110/127, 86.6%), and no differences were found between the two groups.

Ultrasound parameters of the 237 groins are shown in Appendix A. Almost all morphological and dimensional ultrasound parameters significantly differed between metastatic and non-metastatic groups.

### 3.3. Predictive Performances of Ultrasound Variables, Subjective Assessment and Morphonode–RFC

The predictive performances of the ultrasound variables, subjective assessment and Morphonode–RFC are shown in Figure 2. Among ultrasound variables, cortical thickness and short axis diameter demonstrated the highest AUCs (79.3%, 95%CI 73.3–85.4 and 79.1%, 95%CI 72.5–85.8, respectively) and NPVs (85.0%, 95%CI 78.6–90.7, for both of them). The sensitivity and specificity of cortical thickness and short axis were: 72.3% (95%CI 61.4–81.9) and 73.4% (95%CI 66.5–80.3), 71.8% (95%CI 59.7–80.8) and 73.3% (95%CI 66.3–79.8), respectively.

Subjective assessment of the ultrasound examiner and the combination of subjective assessment and cytological examination (FOA, final overall assessment) had: AUC 86.2% (95%CI 80.9–91.5) and 86.7% (95%CI 81.2–91.8), respectively; NPV 92.7% (95%CI 86.7–96.3) and 87.3% (95%CI 81.6–92.0), respectively; sensitivity 88.0% (95%CI 79.0–93.9) and 70.8% (95%CI 59.4–80.7), respectively; and specificity 71.0% (95%CI 63.5–77.6) and 92.9% (95%CI 87.8–96.2), respectively.

The Morphonode–RFC showed the highest performance, compared to that of single ultrasound parameters and subjective assessment, with AUC 91.8% (95%CI 83.2–99.5), NPV 97.1% (95%CI 83.8–100.0), sensitivity 93.3% (95%CI 61.3–100.0), and specificity 92.9% (95%CI 75.0–100.0).

All the described ultrasound variables were included in the Morphonode Predictive Model, except two (medulla thickness and long axis), based on their low contribution to the classification process. The input variable priority of ultrasound parameters, defined according to the Morphonode Predictive Model ranking, is shown in Table 2. Short axis and cortical thickness were excellent outcome predictors and good risk predictors for lymph node metastases; hence their priority was “necessary”. The absence of a nodal core sign or the presence of a perinodal hyperechogenic ring or cortical interruption were fair outcome predictors and excellent risk predictors. Therefore, their priority was “very high”. The other six ultrasound parameters (echogenicity, focal intranodal deposit, vascular flow localization, cortical thickening, vascular flow architecture pattern, cortical–medullar interface distortion) were “high” priority, being fair-good outcome predictors and good risk predictors. The Morphonode Predictive Model can work even if some ultrasound variables are not entered, and its performance could vary depending on the priority of the missing variables.

### 3.4. Malignancy Risk Thresholds and Morphonode–RBM Performances

The two optimality criteria used to define malignancy risk groups yielded two risk thresholds: 0.23 (maximum F1 score) and 0.29 (maximum sensitivity and specificity). The risk groups were defined accordingly as low-risk (*p* < 0.23), moderate-risk (0.23 ≤ *p* ≤ 0.29), and high-risk (*p* > 0.29).

When compared to the observed phenotype, these thresholds lead to a high-performance RBM-based predictor: accuracy 92% (CI95%: 0.87, 0.94), F1 score 87% (CI95%: 0.81, 0.92), sensitivity 0.92% (CI95%: 0.84, 0.97), specificity 0.91% (CI95%: 0.87, 0.95), PPV 0.93% (CI95%: 0.73, 0.90), NPV 0.96% (CI95%: 0.92, 0.99), LR+ = 10.65 (CI95%: 6.64, 18.81), LR− = 0.09 (CI95%: 0.03, 0.18), AUC 0.92% (CI95%: 0.88, 0.96).

### 3.5. Morphonode–DT and Risk Signatures

The Morphonode–DT included four different signatures associated with increased risk levels (Figure 4). The first signature (Metastatic Signature–MET) was verified when at least one of the following three ultrasound parameters (metastatic markers) were present: absence of a nodal core sign, presence of perinodal hyperechogenic ring and cortical interruption. When at least one of these three markers was present, the expected point risk of malignancy was > 86%, and it reached approximately 100% if at least two of them were present.

In the absence of all these three markers, the decisional flow gradually followed the other parameters in order of priority to define the remaining three signatures: High Metastatic Risk (HMR), Moderate Metastatic Risk (MMR) and Low Metastatic Risk (LMR). Each of these three signatures corresponds to an expected point risk (95% bootstrap confidence interval) of 81% (52–90%), 16% (6–25%) and 4% (0–10%), respectively.

Moreover, for MMR, it is possible to describe a series of additional parameters and defined covariates, including vascular flow architecture pattern, vascular flow localization, cortical–medullar interface distortion, cortical thickness, echogenicity, focal intranodal deposits, shape, grouping and color score. In the presence of at least two of them, the expected point risk rises up to 55% (46–64%).

Furthermore, MET is associated with a higher frequency of a greater number of metastatic nodes (Figure 5) than HMR (45.7% vs. 14.3%; MET-HMR proportion difference: 31.4%, 95% CI: 7.1–55.6%, *p*-value: 0.0272).

The following diagram shows the decisional process defining the metastatic risk signatures. The presence of at least one metastatic marker (perinodal hyperechogenic ring, nodal core sign absence, and cortical interruption) leads to a metastatic (MET) signature (red path). The presence of exactly one metastatic marker implies a malignancy risk between 0.86 and 0.9, whereas more than one metastatic marker will lead to a malignant phenotype. Alternatively, a patient might show a high metastatic risk (HMR; orange path), a moderate metastatic risk (MMR; green path), or a low metastatic risk (LMR; blue path) signature. Each box reports the risk estimate (95% confidence interval) associated with the respective signature, measured as the posterior probability of malignancy given the signature. The MMR signature can be further supported by the presence of at least two diagnostic covariates, increasing the malignancy risk estimate from 0.16 to 0.55.

Diagnostic covariates (variable, range of positivity, description):

Vascular flow architecture pattern: 2–4 (2 = Scattered, 3 = Branched, 4 = Chaotic);

Vascular flow localization: 2–4 (2 = Peripheral, 3 = Extranodal, 4 = Combined);

Cortical–medullar interface distortion CMID: 2–3 (2 = Focal, 3 = Diffused, 4 = Medulla not visible);

Cortical thickening: 2–4 (2 = focal, 3 = concentric, 4 = eccentric);

Echogenicity: 2 (Inhomogeneous);

Focal intranodal deposit: 1–3 (1 = Hyperechoic, 2 = Anaechoic (cystic areas), 3 = both);

Shape: 3 (irregular);

Grouping: 2–3 (2 = moderate, 3 = complete);

Color score: 3–4.

Signatures, from left to right, are ordered by decreasing malignancy risk and proportion of multiple metastases (MET: metastatic signature, HMR: high metastatic risk, MMR: moderate metastatic risk, and LMR: low metastatic risk). The y-axis measures the number of groins, while the counts above each bar show the frequency of non-malignant [0], malignant with a single positive lymph node [1], and malignant with multiple positive lymph nodes [>1]. Bars are shaded according to the proportion of positive lymph nodes. Although MET and HMR signatures show the same proportion of positive groins, the former shows a significantly higher proportion of multiple metastatic lymph nodes compared to the latter (MET-HMR proportion difference: 31.4%, 95% CI: 7.1–55.6%, and *p*-value: 0.0272).

### 3.6. Prediction Error

We defined a reference cut-off of one to define the difference between the predicted value made by the model and the real value. In fact, a value below one directly correlates with a prediction that can be considered appropriate. When this output shows a result higher than one, it is an alert for the user that the prediction should not be considered feasible on the base of the inserted ultrasound parameters.

### 3.7. Similarity Search Module

We validated the similarity search module against the observed phenotype by assigning the majority phenotype of the top five similar profiles to the predicted outcome. We obtained the following performances: accuracy = 81% (CI95%: 0.73, 0.87), F1 score = 0.82 82%(CI95%: 0.73, 0.87), sensitivity = 88%(CI95%: 0.77, 0.95), specificity = 75% (CI95%: 0.63, 0.85), PPV = 77% (CI95%: 0.65, 0.86), NPV = 87% (CI95%: 0.75, 0.94), LR+ = 3.56 (CI95%: 2.36, 5.90), LR− = 0.17 (CI95%: 0.07, 0.32), and AUC = 82% (CI95%: 0.72, 0.88).

All the modules included in the Morphonode Predictive Model are illustrated in Figure 6:-Stratification into malignant or benign (Morphonode–RFC)-Point risk estimation (Morphonode–RBM)-Risk signature (Morphonode–DT)-Estimated prediction error of the combination of the three modules RFC, RMB and DT-Top five similar profiles (Morphonode–SP).

**Figure 6 cancers-15-01121-f006:**
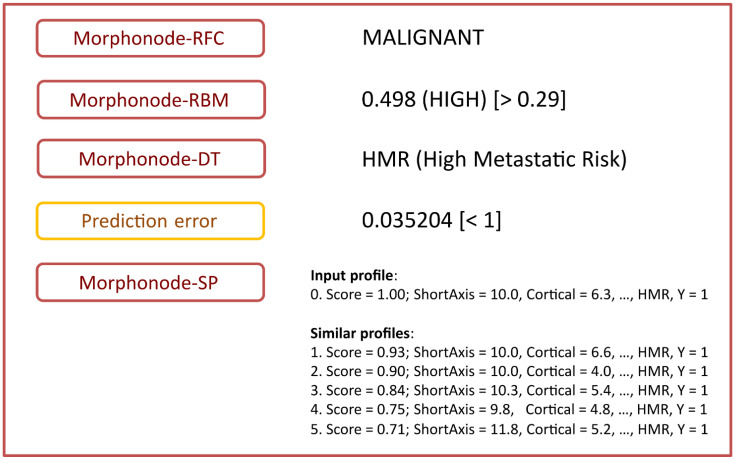
Morphonode Predictive Model output scheme.

The Morphonode Predictive Model output includes the RFC-based diagnosis (Morphonode–RFC), the malignancy risk signature prediction (one among MET, HMR, MMR, and LMR), an estimation of malignancy-risk interpretable as low (<0.23), moderate (between 0.23 and 0.29), and high (>0.29) and a prediction error related to these three modules. Finally, the top five similar profiles to the input are retrieved from the Morphonode dataset, and their profile is shown by decreasing similarity scores.

## 4. Discussion

The Morphonode model provided four output modules: (i) binary malignancy prediction (Morphonode–RFC), (ii) point malignancy risk estimation (Morphonode–RBM), (iii) risk signature (Morphonode–DT), and (iv) selection of the top five similar profiles in the study series (Morphonode–SP).

Morphonode–RFC predicted metastatic lymph nodes with an accuracy of 93.3%, a sensitivity of 93.3%, a specificity of 92.9%, and an NPV of 97.1%. This performance was higher than the subjective assessment of expert examiners. Morphonode–DT identified four specific signatures correlated with the risk of metastases: metastatic signature (MET), high metastatic risk (HMR), moderate metastatic risk (MMR) and low metastatic risk (LMR). The point risk of each signature was 100%, 81%, 16%, and 4%, respectively. MET signature also correlated with the risk of finding more than one metastatic lymph node (45.7%). Unless the prediction error is below the threshold (<1), these modules allow the practitioners to generate a single and reliable diagnosis by using a simple majority-wins criterion, where a positive (negative) diagnosis is achieved if two out of three modules yield a malignant (non-malignant) prediction (Figure 7). Although rare, two exceptions are possible: (i) both RBM and DT return an intermediate (moderate risk) prediction; (ii) two modules yield opposite predictions, and the third one returns an intermediate prediction. In these cases, a reliable diagnosis is not possible, and it is advisable to perform a biopsy.

The scheme below schematically shows how the Morphonode Predictive Model assigns a subject to a given class, including metastatic (red and orange boxes), non-metastatic (blue boxes), and moderate (undefined, green boxes). Morphonode Predictive Model modules are reported in white boxes. The model defines three malignancy risk levels (high, moderate, and low) and four metastatic risk signatures (metastatic, high-risk, moderate-risk, and low-risk). Three modules, namely the random forest classifier (Morphonode–RFC), the robust binomial model (Morphonode–RBM), and the decision tree (Morphonode–DT), determine which class the patient belongs to. An estimate of malignancy risk is given by both the RFC (binary value 0/1) and the RBM (continuous value from 0 to 1), while the DT module outputs a signature associated with the predicted number of metastases: LMR (0), MMR (uncertain, 0 or 1), HMR (1), MET (>1).

To apply the Morphonode model in clinical practice, we suggest following the workflow diagram shown in Figure 8. Briefly, if more than one of the three metastatic markers is present, the studied node(s) should be considered metastatic with a probability of nearly 100% (MET signature). We also recommend applying the open-source R package Morphonode Predictive Model, providing comprehensive information about an automated diagnosis, estimated prediction error, risk evaluation and similarity profiling, in addition to risk signatures (R package available at: https://github.com/Morphonodepredictivemodel, accessed on 23 November 2022). Otherwise, in case the open source R package is not available, the Morphonode–DT can be manually applied, with a reliable prediction in case of HMR or LMR signatures.

The scheme below shows the conceptual workflow followed by the Morphonode Predictive Model. The arrows show the priority that should be given to the software output. If one or more metastatic markers (perinodal hyperechogenic ring, nodal core sign absence, and cortical interruption) are present, the associated signature is MET, and the diagnosis is malignant (this evidence can be further supported by the RFC decision). Otherwise, the Morphonode–DT module will choose among LMR, MMR, or HMR signatures. In these cases, the output of the Morphonode–RFC module will be more important for the decision process. Simultaneously, the Morphonode–RBM and Morphonode–SP modules will give an ultrasound-associated metastatic risk estimate and a similarity profiling, supporting the clinical decision, especially in case of difficult classifications.

The strengths of this study are the prospective design, the use of a validated dataset, and the inclusion of surgically removed lymph nodes with a known histological outcome. All the ultrasound examinations were performed by experienced examiners at a high-volume oncological referral center with a predefined registration protocol. Moreover, we were able to define a rigorous procedure for validating the predictive model and results. A limitation is that patients included in this study were recruited from a single institution, and the proposed model currently has not yet been externally validated.

To our knowledge, there are no similar studies on artificial intelligence applied to ultrasound in gynecologic oncology reported in the literature. So far, only a few authors have described more generally the application of artificial intelligence to the diagnosis of gynecologic cancers [19,20,21]. In our institution, we developed and validated radiomics models, applied to ultrasound images, capable of differentiating high-risk endometrial cancer from other endometrial cancers (low-, intermediate- and high-intermediate risk) [19]. Other authors applied deep neural network analysis to ultrasound images to discriminate between benign and malignant ovarian tumors [20]. They demonstrated that ultrasound image analysis using deep neural networks could predict ovarian malignancy with a diagnostic accuracy comparable to that of human expert examiners. Chiappa and colleagues [21] developed machine learning models after extracting a large number of radiomics features from ultrasound images to predict the risk of malignancy of a uterine mesenchymal lesion. Ultrasound-based radiomics models showed good accuracy in discriminating between myomas and sarcomas, and the authors concluded that radiomics analysis could represent a decision-support tool in the identification of sarcomas, which are often difficult to diagnose. Finally, a single study from our working group explored the radiomic features of PET/CT images from the primary tumor site of vulvar tumors, showing poor results in the prediction of tumor biology and prognosis [22]. Regarding ultrasound and inguinal lymph nodes, there are no existing studies using artificial intelligence to predict metastases. De Gregorio et al. in 2013 [23] described the performance of ultrasound examination in discriminating between metastatic and non-metastatic lymph nodes by considering the following features: absence of fatty hilum, irregular shape, cortical region diameter and vascularization pattern. Ultrasound performed well, with a PPV of 82.9% and an NPV of 87.5%. More recently, in 2018, Pouwer et al. [24] identified the following ultrasound parameters indicative of metastatic lymph nodes: short-axis ≥10 mm in oval lymph nodes or short-axis ≥8 mm in circular-shaped lymph nodes when associated with malignant characteristics (i.e., hilar hypoechogenicity, general attenuation, irregular borders, or abnormal vascular pattern). In this study, the PPV was 6.8%, and the NPV was 100% [24].

Artificial intelligence was applied for the evolution of axillary nodes in patients with breast cancer, with similar or better performance than baseline imaging [25,26].

In 2021, Yu et al. developed a multiomic signature based on clinical, pathologic and molecular characteristics, axillary lymph node and tumor radiomic features selected from preoperative MR images. This model, based on the random forest machine learning technique, achieved a good diagnostic performance in identifying metastatic lymph nodes with an AUC of 90% [27].

Similarly, in the same year, Zhang et al. constructed a nomogram using MRI radiomic signature with an AUC of 81% [28].

Moreover, deep learning models based on CT scan imaging were developed and validated to predict cervical node diagnosis in patients with thyroid cancer, with an AUC ranging from 78% to 88% [29].

Finally, a machine-learning-based radiomics model applied to PET/CT images was recently developed by Song et al. to predict axillary node metastases. This model showed an accuracy of 80% [30].

The results of this study provide a preoperative diagnostic model able to reliably identify metastatic lymph nodes in patients with vulvar cancer. That is clinically relevant to address treatment strategies, personalize treatments, schedule the operating room and select surgical team and equipment [5,13,31,32,33,34,35,36,37,38,39]. In particular, the aim of modern vulvar cancer management is to minimize treatment-related morbidity without compromising survival [31,32,33,34,35,36,37,38,39]. Sentinel lymph node biopsy—introduced in order to reduce morbidity, is currently applicable in strictly selected patients with early-stage disease and 572 specific clinical characteristics, including unifocal primary lesion, tumor size < 4 cm and preoperative node negative evaluation [40,41].

These strict selection criteria are necessary to avoid the risk of large undiagnosed metastases that have the potential to divert tracer drainage to falsely negative lymph nodes, increasing the risk of missed metastases with negative prognostic impact [3,4,7,42,43,44,45]. The present study demonstrated that the Morphonode Predictive Model is able to predict inguinal negativity, suitable for sentinel node biopsy, with a low rate of false negative cases, and to correctly identify node-positive patients who would benefit from inguinal lymphadenectomy, minimizing the number of false positive cases. Therefore, the Morphonode model could lay the ground for future studies aiming to expand indications for minimally invasive surgery, as well as the criteria for sentinel node procedures, for further supporting the results of the GroSNaPET study [39].

Another observation can be made about the choice of many reference centers to omit the frozen section of the sentinel lymph node in favor of definitive pathology in order to avoid the loss of micrometastases and to improve the measurement of macrometastases. Obviously, this practice carries a risk of second surgeries corresponding to the rate of metastases in the sentinel lymph nodes, which averages around 21–35.8% in the literature [44].

Therefore, we could argue that by improving the preoperative detection of lymph node metastases, the Morphonode model could potentially help to reduce the rate of positive sentinel lymph nodes at definitive pathology and, consequently, the rate of second radical surgeries.

Finally, the possibility of empowering the predictive capacity of ultrasound examination, even in the hands of less-trained operators, through a widely available and low-cost tool that can also be employed directly at the first outpatient visit is something that could further support the routine use of ultrasound in the preoperative workup process. In fact, the role of ultrasound is not yet globally recognized: for example, the current National Comprehensive Cancer Network (NCCN) guidelines [13] do not mention ultrasound among workup examinations at all, except in the one section specifically for vulvar melanoma.

On the other hand, the updated European Society of Gynecological Oncology (ESGO) guidelines [45] clearly recommend the use of inguinal ultrasound in the preoperative workup, but it is only limited to experienced operators: as a first step in preoperative staging, in patients considered for sentinel lymph node biopsy, in order to identify the presence of non-palpable macrometastases; and also as a second step in cases of suspicious inguinal lymph nodes on imaging, as a guide for fine-needle aspiration or core biopsy.

Since the Morphonode model has demonstrated a diagnostic performance even superior to the subjective assessment of experienced ultrasound examiners, we can postulate that in the near future, it could be employed to support nonexpert examiners in achieving high predictivity on lymph node assessment by ultrasound in different settings. Further studies could include other clinical parameters, such as “HPV association”, in the model to improve its performance.

## 5. Conclusions

In conclusion, the Morphonode Predictive Model was demonstrated to be an accurate diagnostic method for predicting inguinal lymph node metastases. Multicenter prospective studies are necessary to externally validate it on a larger population. In the future, a further integrated model could be developed, including clinical parameters, pathological data and radiomics features.

## Figures and Tables

**Figure 1 cancers-15-01121-f001:**
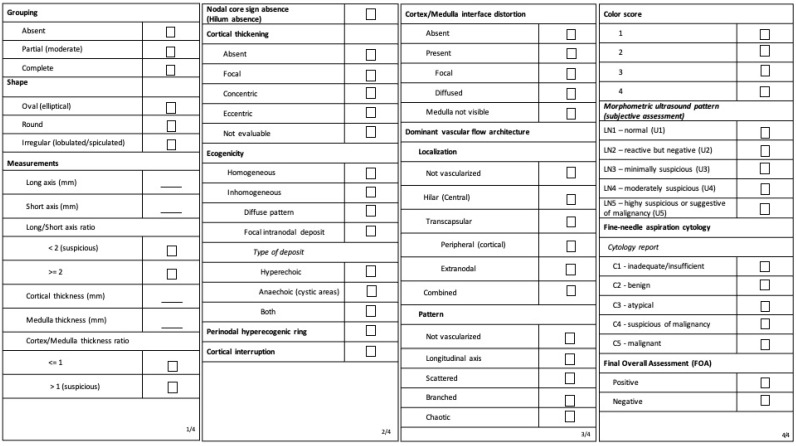
Ultrasound features registration form. All features are described according to VITA nomenclature. The corresponding terms previously used in the Morphonode Study are reported in brackets.

**Figure 2 cancers-15-01121-f002:**
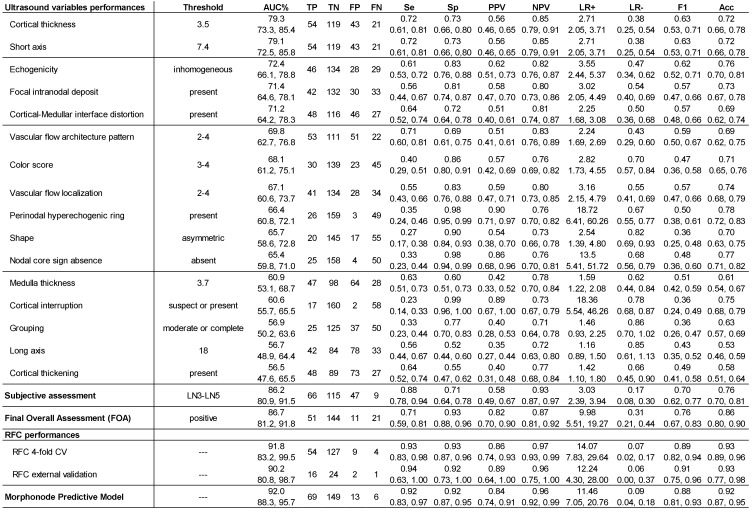
Predictive performances of ultrasound features, subjective assessment and Morphonode–RFC. Ultrasound variables are grouped in four blocks according to the Area Under the ROC Curve: AUC > 75% (first), AUC > 70% (second), AUC > 65% (third), and AUC <= 65% (fourth). The optimal cut-point value (threshold) was defined by a max-Sensitivity/max-Specificity criterion. AUCs, performance indices, and their 95% confidence intervals are reported. Performance indices acronyms for ultrasound variables, subjective assessment, and Morphonode–RFC are the following: TP, True Positives; TN, True Negatives; FP, False Positives; FN, False Negatives; Se, Sensitivity; Sp, Specificity; PPV, Positive Predictive Value (Precision); NPV, Negative Predictive Value; LR+, Positive Likelihood Ratio, LR−, Negative Likelihood Ratio; F1, F1 score; Acc, Accuracy. The F1 score is defined as the harmonic mean between precision (PPV) and sensitivity (Se), thus penalizing huge differences between sensitivity and specificity. This provides a more reliable predictive accuracy measure than Acc = (TP + TN)/(TP + TN + FP + FN).

**Figure 3 cancers-15-01121-f003:**
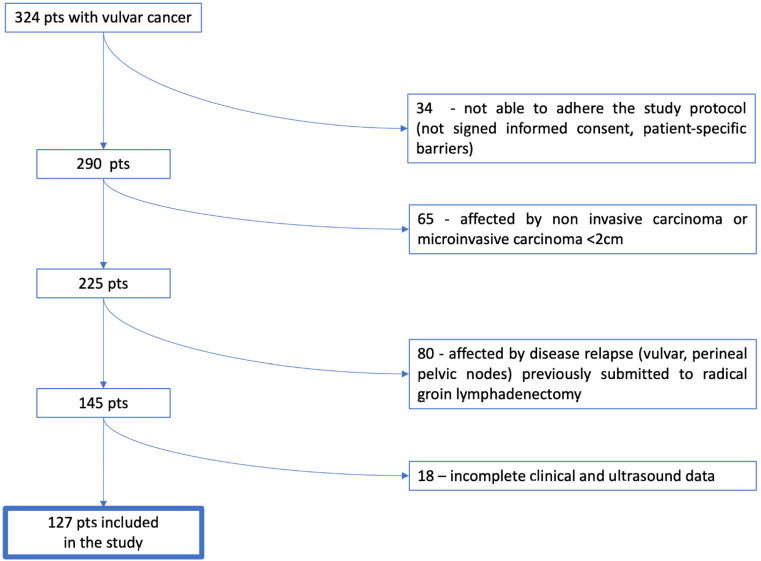
Flow chart: patients’ selection.

**Figure 4 cancers-15-01121-f004:**
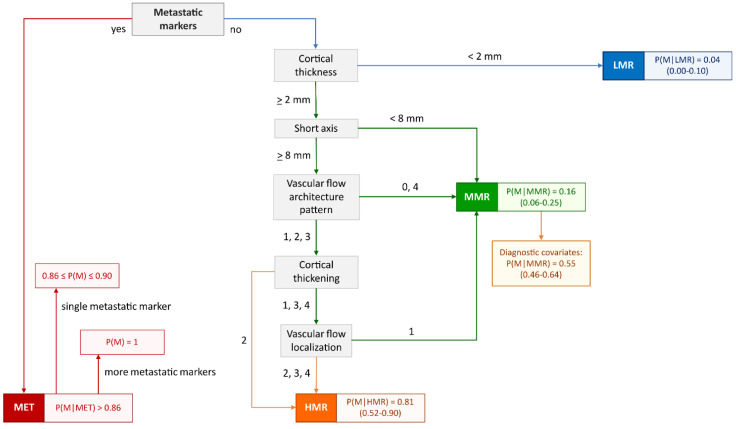
Decision tree flow chart: risk signatures.

**Figure 5 cancers-15-01121-f005:**
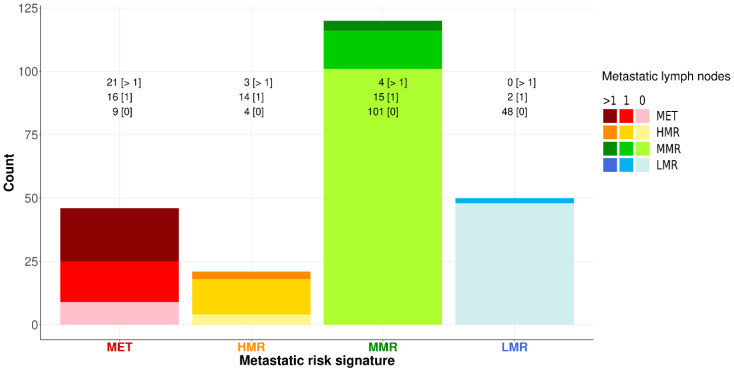
Metastasis risk signatures (MRSs) and the number of positive nodes.

**Figure 7 cancers-15-01121-f007:**
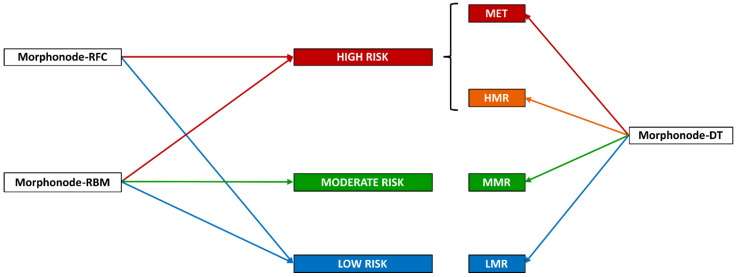
Morphonode Predictive Model workflow.

**Figure 8 cancers-15-01121-f008:**
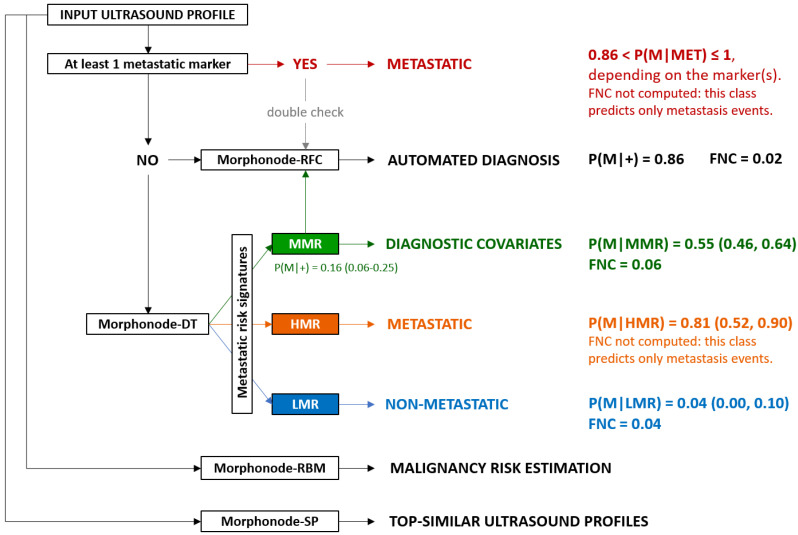
Morphonode Predictive Model workflow.

**Table 1 cancers-15-01121-t001:** Clinical, surgical, histopathologic and ultrasound features.

Characteristic	All (n = 127)	N0—Negative Lymph Node at Histology (n = 71)	N1—Positive Lymph Node at Histology (n = 56)	Test [a]	Estimate [b]	CI 95% [c]	*p*-Value
Age (years): median (range)	69 (32–95)	71 (44–95)	68 (32–89)	Wilcoxon	−2	−6, 4	0.402
**Tumor site ***							
Anterior	45/127 (35.4%)	22	23	proportions	0.411, 0.310	−0.083, 0.285	0.321
Lateral	61/127 (48%)	34	27	proportions	0.482, 0.479	−0.175, 0.182	0.999
Posterior	15/127 (11.8%)	13	2	proportions	0.036, 0.183	−0.266, −0.029	0.023
Central	5/127 (4%)	2	3	proportions	0.054, 0.028	−0.061, 0.112	0.786
Absent	1/127 (0.8%)	0	1	proportions	0.018, 0.000	−0.033, 0.069	0.905
**Focality**							
Unifocal	106/127 (83.5%)	63	43	proportions	0.768, 0.887	−0.268, 0.029	0.119
Multifocal	21/127 (16.5%)	8	13	proportions	0.232, 0.113	−0.029, 0.268	0.119
**Type of vulvar surgery**							
Partial	66/127 (51%)	24	42	proportions	0.750, 0.338	0.238, 0.586	<0.001
Radical	61/127 (49%)	29	32	proportions	0.571, 0.408	−0.026, 0.352	0.099
**Side of surgery**							
Monolateral	16/127 (13%)	12	4	proportions	0.071, 0.169	−0.224, 0.029	0.169
Bilateral	111/127 (87%)	59	52	proportions	0.929, 0.831	−0.029, 0.224	0.169
**Histotype**							
Squamous	110/127 (86.6%)	63	46	proportions	0.821, 0.887	−0.206, 0.074	0.423
Paget	9/127 (7%)	5	4	proportions	0.071, 0.070	−0.090, 0.092	0.999
Melanoma	5/127 (4%)	2	3	proportions	0.054, 0.028	−0.061, 0.112	0.786
Basocellular	1/127 (0.8%)	0	1	proportions	0.018, 0.000	−0.033, 0.069	0.905
Adenocarcinoma	1/127 (0.8%)	1	0	proportions	0.000, 0.014	−0.056, 0.027	0.999
Sarcoma	1/127 (0.8%)	0	1	proportions	0.018, 0.000	−0.033, 0.069	0.905
**Maximum tumor diameter** (mm)							
Median (range)	30 (2–160)	25.5 (0–100)	41 (4–90)	Wilcoxon	12	5, 20	<0.001
<20 mm	28/122 (23%)	22	6	proportions	0.107, 0.310	−0.353, −0.052	0.012
20–40 mm	52/122 (42.6%)	31	21	proportions	0.375, 0.437	−0.249, 0.126	0.604
>40 mm	42/122 (34.4%)	15	27	proportions	0.482, 0.211	0.093, 0.449	0.002
**Grading** (squamous histotype)							
G1	20/110	14	6	proportions	0.107, 0.197	−0.229, 0.049	0.255
G2	63/110	34	29	proportions	0.518, 0.479	−0.152, 0.230	0.797
G3	19/110	7	12	proportions	0.214, 0.099	−0.028, 0.260	0.118
**Depth of invasion** ° (squamous histotype; mm)							
Median (range)	6 (0–19)	5 (0–12)	6 (0.9–19)	Wilcoxon	2	0, 3	0.020
<5 mm	32	20	12	proportions	0.214, 0.282	−0.233, 0.099	0.507
>=5 mm	54	23	31	proportions	0.554, 0.324	0.044, 0.415	0.016
**Lymphovascular invasion**							
Absent	31/127 (24%)	22	9	proportions	0.161, 0.310	−0.309, 0.011	0.083
Present	96/127 (76%)	49	47	proportions	0.839, 0.690	−0.011, 0.309	0.083
**Stage**							
IB	59 (46.5)	58	1	proportions	0.018, 0.817	−0.911, −0.687	<0.001
II	4 (3.2)	4	0	proportions	0.000, 0.056	−0.126, 0.013	0.196
III	40 (31.5)	0	40	proportions	0.714, 0.000	0.580, 0.849	<0.001
IV	4 (3.2)	0	4	proportions	0.071, 0.000	−0.012, 0.155	0.076
Relapse	12 (9.4)	5	7	proportions	0.125, 0.070	−0.066, 0.176	0.460
Post-RT/CT	8 (6.3)	4	4	proportions	0.071, 0.056	−0.086, 0.116	0.999

Results are presented as number (%) or median (range). Age, tumor site, and focality are clinical parameters. The type and side of surgery are surgical parameters. Histotype, maximum tumor diameter, grading, depth of invasion, lymphovascular invasion and stage are histological parameters. [a] Two-sided Wilcoxon’s rank sum test for continuous variables; Two-sided test for equality of proportions for count variable. [b] Either distribution shift estimate (Wilcoxon’s rank sum test) or proportions for positive and negative lymph nodes (proportions test). [c] 95% confidence interval for either shift estimate (Wilcoxon’s rank sum test) or positive–negative proportion difference (proportions test). * Tumor site at clinical examination. ° Depth of invasion at histology.

**Table 2 cancers-15-01121-t002:** Input variable priority.

Variable	RFC Ranking	Discriminant	Metastatic Risk	Priority	Description
Short axis (mm)	*	+++	++	Necessary	Excellent outcome predictor, good risk predictor
Cortical thickness (mm)	*	+++	++	Necessary	Excellent outcome predictor, good risk predictor
Nodal core sign absence	+++	+	*	Very high	Fair outcome predictor, excellent risk predictor
Perinodal hyperecogenic ring	++	+	*	Very high	Fair outcome predictor, excellent risk predictor
cortical interruption	++	−	*	Very high	Fair outcome predictor, excellent risk predictor
Echogenicity	+++	++	+++	High	Good outcome predictor, good risk predictor
Focal intranodal deposit	+++	++	++	High	Good outcome predictor, good risk predictor
Vascular flow localization	+++	+	−	High	Good outcome predictor
Cortical thickening	+++	−	+	High	Good outcome predictor
Vascular flow architecture pattern	++	+	++	High	Fair outcome predictor, good risk predictor
Cortical–medullar interface distortion	++	++	++	High	Fair outcome predictor, good risk predictor
Shape	+	+	++	Low	Poorly informative
Grouping	+	−	+	Low	Poorly informative
Color score	+	+	−	Low	Poorly informative
Medulla (mm)	−	−	−	Unnecessary	Not informative
Long axis (mm)	−	−	−	Unnecessary	Not informative

The Morphonode Predictive Model is robust to missing data. However, the higher the contribution of a variable to the predictive accuracy, the higher the contribution to the prediction error given by its absence. The table shows the priority of each ultrasound feature on the basis of its RFC ranking (i.e., the average of min-max-normalized MDA and MDG indices), its discriminant accuracy in terms of AUC% (see Figure 2 for details), and its associated metastatic risk (calculated as P(M|+) = positive predictive value). Variables with “necessary” or “very high” should never be missing, given their critical role in the prediction process. All the other variables have a moderate-to-null effect over prediction accuracy. In addition, the software provides an imputed value based on either LASSO regression (continuous variables) or decision trees (categorical variables). RFC ranking: * top/stable ranking; +++ > 20% of the top-ranked; ++ > 5% of the top-ranked; + ≤ 5% of the top-ranked; − Not informative. Discriminant (individual discriminant accuracy as AUC): +++ > 75%; ++ > 70%; + > 65%; − ≤ 65%. Individual metastatic risk prediction (positive predictive value): * ≥ 0.8; +++ ≥ 0.6; ++ ≥ 0.5; + ≥ 0.4; − < 0.4.

## Data Availability

The data presented in this study are available on reasonable request from the corresponding author.

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
