# Peer review of "Evaluating the Risk of Inguinal Lymph Node Metastases before Surgery Using the Morphonode Predictive Model: A Prospective Diagnostic Study in Vulvar Cancer Patients"

_cancers, 2023, doi:10.3390/cancers15041121_

Round 1

Reviewer 1 Report

Dear authors,

congratulations for the impressive work. This is a further assurance for ever improving care for vulvar cancer patients. 

The significance of an algorithm for stratifying nodal metastasis risk is essential to enter the new era for vulvar cancer management, and your work is an essential step towards this direction.

I only have a minor consideration: HPV positivity (p16 positivity as a proxy) could be included in the criteria? and if it was evaluated, was it significant?

Author Response

Point 1: I only have a minor consideration: HPV positivity (p16 positivity as a proxy) could be included in the criteria? and if it was evaluated, was it significant?

Response 1: Thank you for your comment. We agree with the observation: it can be clinically relevant to know if the tumor is HPV associated or HPV independent and to include this parameter in the model. However, in the present study, we develop an ultrasound-based-model including only ultrasound parameters to discriminate between metastatic and non-metastatic inguinal lymph nodes. Further studies could include other clinical parameters, such as “HPV association” in the model to improve its performance. A comment has been added in the discussion section. See lines 630-632.

Reviewer 2 Report

The manuscript by Fragomeni et al titled "Evaluating the risk of inguinal lymph node metastases before 2 surgery using the MORPHONODE PREDICTIVE MODEL: a 3 prospective diagnostic study in vulvar cancer patients" is a prospective study aimed  to build a robust, multi-modular model based on machine learning, to discriminate between metastatic and non-metastatic inguinal lymph nodes in patients with vulvar cancer. The study is very interesting and innovative. The introduction is clear and exhaustive, the aim is clearly reported. The methodology is of high quality and well-described. The results are very relevant and they are well supported by the tables and figures. The Discussion is complete and clearly explain the innovations and the limitations of the study, as well as the clinical implications of the findings.

Author Response

Thank you for your comment. 

Reviewer 3 Report

Overall the manuscript is well written and well researched.  It will be of great interest to gynecology oncology surgeons who may use this to guide management of women with a diagnosis of vulval carcinoma. 

I have only a few suggestions for consideration:

1.  In the Simple Summary:

Line 34 "Inguinal nodes status represents one a key element in defining prognosis and treatment strategies in vulvar cancer patients." 

Could the sentence read - Inguinal nodes status represents one of the key elements in defining prognosis and treatment strategies in vulvar cancer patients.

2.  Line 36 " Several imaging methods are currently recommended in the guidelines (CT, PET/CT, MRI, US) basing on performance data that are still not conclusive in the literature."

Could the sentence read - Several imaging methods are currently recommended in the guidelines (CT, PET/CT, MRI, US) based on performance data that are still not conclusive in the literature.

3.  In Standard of reference:

Line 432 - "All patients underwent surgery on primary tumor and inguinal lymph nodes, according to international guidelines[11,15]."

Could the sentence read - All patients underwent surgery on the primary tumor and inguinal lymph nodes, according to international guidelines [11,15].

In the Discussion:

Line 571 - "Sentinel lymph node biopsy - introduced in order to reduce morbidity is currently applicable in strictly selected patients with early stage disease and specific clinical characteristics, including unifocal primary lesion, tumor size < 4 cm, pre-operative 573 node negative evaluation [42-43].

Could the sentence read - Sentinel lymph node biopsy - introduced in order to reduce morbidity is currently applicable in strictly selected patients with early stage disease and specific 572 clinical characteristics, including unifocal primary lesion, tumor size < 4 cm and pre-operative node negative evaluation [42-43].

The Standard of Reference is described well. 

In the discussion you make an interesting point in line 585.  Many reference centers, like the one I work for, do not perform frozen sections on sentinel lymph nodes for vulval carcinomas, in  order to avoid missing a  micrometastasis.  Some pathology practices however do perform imprint cytology for their intra-operative assessment to guide the need for a subsequent lymph node dissection in the same operation, without compromising on lost tissue.

As vulval squamous cell carcinomas are the most type of vulval cancers it is suggested that the risk for nodal metastases is also now dependent on if the tumor is HPV associated or HPV independent.

Author Response

Point 1: In the Simple Summary:

Line 34 "Inguinal nodes status represents one a key element in defining prognosis and treatment strategies in vulvar cancer patients."

Could the sentence read - Inguinal nodes status represents one of the key elements in defining prognosis and treatment strategies in vulvar cancer patients.

Response 1: We modified the sentence accordingly. See lines 34-35.

Point 2: Line 36 " Several imaging methods are currently recommended in the guidelines (CT, PET/CT, MRI, US) basing on performance data that are still not conclusive in the literature."

Could the sentence read - Several imaging methods are currently recommended in the guidelines (CT, PET/CT, MRI, US) based on performance data that are still not conclusive in the literature.

Response 2: We modified the sentence accordingly. See lines 36-37.

Point 3.  In Standard of reference:

Line 432 - "All patients underwent surgery on primary tumor and inguinal lymph nodes, according to international guidelines[11,15]."

Could the sentence read - All patients underwent surgery on the primary tumor and inguinal lymph nodes, according to international guidelines [11,15].

Response 3: We modified the sentence accordingly. See lines 151-152.

Point 4: In the Discussion:

Line 571 - "Sentinel lymph node biopsy - introduced in order to reduce morbidity is currently applicable in strictly selected patients with early stage disease and specific clinical characteristics, including unifocal primary lesion, tumor size < 4 cm, pre-operative 573 node negative evaluation [42-43].

Could the sentence read - Sentinel lymph node biopsy - introduced in order to reduce morbidity is currently applicable in strictly selected patients with early stage disease and specific 572 clinical characteristics, including unifocal primary lesion, tumor size < 4 cm and pre-operative node negative evaluation [42-43].

Response 4: We modified the sentence accordingly. See lines 582-585.

Point 5: The Standard of Reference is described well.

In the discussion you make an interesting point in line 585.  Many reference centers, like the one I work for, do not perform frozen sections on sentinel lymph nodes for vulval carcinomas, in  order to avoid missing a  micrometastasis.  Some pathology practices however do perform imprint cytology for their intra-operative assessment to guide the need for a subsequent lymph node dissection in the same operation, without compromising on lost tissue.

As vulval squamous cell carcinomas are the most type of vulval cancers it is suggested that the risk for nodal metastases is also now dependent on if the tumor is HPV associated or HPV independent.

Response 5: Thank you for your comment. We agree with the observation: it can be clinically relevant to know if the tumor is HPV associated or HPV independent and to include this parameter in the model. However, in the present study, we develop an ultrasound-based-model including only ultrasound parameters to discriminate between metastatic and non-metastatic inguinal lymph nodes. Further studies could include other clinical parameters, such as “HPV association” in the model to improve its performance. A comment has been added in the discussion section. See lines 630-632.
